# COMPARING FIXED AND ADAPTIVE COMPUTATION TIME FOR RECURRENT NEURAL NETWORKS

**Daniel Fojo**[†]**, Víctor Campos**[‡]**, Xavier Giró-i-Nieto**[†]

[†]Universitat Politècnica de Catalunya, [‡]Barcelona Supercomputing Center
daniel.fojo@estudiant.upc.edu, victor.campos@bsc.es, xavier.giro@upc.edu

## 1 INTRODUCTION

Deep networks commonly perform better than shallow ones (Krizhevsky et al., 2012; Simonyan & Zisserman, 2015; He et al., 2016), but allocating the proper amount of computation for each particular input sample remains an open problem. This issue is particularly challenging in sequential tasks, where the required complexity may vary for different tokens in the input sequence. Adaptive Computation Time (ACT) (Graves, 2016) was proposed as a method for dynamically adapting the computation at each step for Recurrent Neural Networks (RNN). ACT introduces two main modifications to the regular RNN formulation: (1) more than one RNN steps may be executed between an input sample is fed to the layer and and this layer generates an output, and (2) this number of steps is dynamically predicted depending on the input token and the hidden state of the network. In our work, we aim at gaining intuition about the contribution of these two factors to the overall performance boost observed when augmenting RNNs with ACT. We design a new baseline, Repeat-RNN, which performs a constant number of RNN state updates larger than one before generating an output. Surprisingly, such uniform distribution of the computational resources matches the performance of ACT in the studied tasks. We hope that this finding motivates new research efforts towards designing RNN architectures that are able to dynamically allocate computational resources. Source code is publicly available at https://imatge-upc.github.io/danifojo-2018-repeatrnn/.

## 2 REPEAT-RNN

An RNN takes an input sequence $\boldsymbol{x} = (x_1, \ldots, x_T)$ and generates a state sequence $\boldsymbol{s} = (s_1, \ldots, s_T)$ by iteratively applying a parametric state transition model $\mathcal{S}$ from $t = 1$ to $T$:

$$s_t = \mathcal{S}(s_{t-1}, x_t) \tag{1}$$

Repeat-RNN can be seen as an ablation of ACT, where the capability of dynamically adapting the number of steps per input token is dropped. This number of steps, which can be understood as the number of repetitions for each input token, is set in Repeat-RNN with a hyperparameter, $\rho$:

$$s_t^n = \begin{cases} \mathcal{S}(s_{t-1}, x_t^1) \text{ if } n = 1 \\ \mathcal{S}(s_t^{n-1}, x_t^n) \text{ if } 1 < n \leq \rho \end{cases} \tag{2}$$

$$s_t = s_t^\rho \tag{3}$$

As in ACT (Graves, 2016), $x_t^n = (\delta_{1,n}, x_t)$ is the input token augmented with a binary flag that indicates whether it is the first time the network sees it. Note that Equation 3 differs from the original ACT formulation, where $s_t$ is the weighted average of all the intermediate states $s_t^n$ weighted by the halting probabilities. Although this may seem unintuitive, we hypothesize that Graves (2016) adopted this solution so that the halting probabilities would appear during the backwards pass. On the other hand, Repeat-RNN simply returns the last state.

As shown in Figure 1, the formulation of Repeat-RNN is equivalent to repeating each element of the input sequence $\rho$ times and adding a binary flag indicating whether the token is new or repeated.

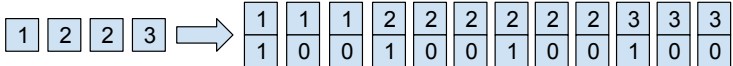

Figure 1: Example of a modified input sequence for Repeat-RNN with $\rho = 3$.

## 3 EXPERIMENTS

We evaluate the models on two tasks already studied by the original ACT paper (Graves, 2016), namely *parity* and *addition*. In the *parity* task, the network must learn how to compute the parity (or XOR) of the input vector, which is given as a single token. Note that the recurrent connection in the RNN is not used in this setup unless the model visits the input token more than once. The *addition* task consists in cumulatively adding the values of a sequence of five numbers, coded by one-hot encodings of their digits. The required output is the sum of all inputs up to the current one, represented as a set of six simultaneous classifications (for the 6 possible digits in the result of the sum). We consider a task as being solved when the accuracy reached 98%. Due to space limitations, more detailed descriptions of the tasks are included in Appendix A. We follow the same experimental setup as Graves (2016), but we do not parallelize SGD with *HogWild!*, and increase the learning rate to $10^{-3}$. The experiments are implemented with TensorFlow and run on a single NVIDIA GPU.

Tables 1 and 2 show a comparison between a simple RNN, ACT and our proposed model, Repeat-RNN. We observe that in our experiments (1) Repeat-RNN learns how to solve the tasks with less SGD training steps than ACT, and (2) it actually solves them with less repetitions than ACT. For example, in Table 1 Repeat-RNN $\rho = 2$ learns to solve the task in only 22k training steps, less than half than the required by the best ACT-RNN configuration $\tau = 10^{-2}$. Notice how $\rho$ is a task-dependent hyperparameter, as the same Repeat-RNN $\rho = 2$ cannot solve the addition task reported in Table 2. In that task, the solution that requires the smallest amount of repetitions is also found with a fixed $\rho = 3$ from Repeat-RNN.

Table 1: Experimental results for the parity task.

| Model | Task solved | Training steps | Average repetitions |
|---|---|---|---|
| RNN | No | - | 1.00 |
| ACT-RNN, $\tau = 10^{-1}$ | No | - | 1.00 |
| ACT-RNN, $\tau = 10^{-2}$ | Yes | 53 k | 1.81 |
| ACT-RNN, $\tau = 5 \cdot 10^{-3}$ | Yes | 356 k | 2.03 |
| ACT-RNN, $\tau = 10^{-3}$ | Yes | 55 k | 2.04 |
| Repeat-RNN, $\rho = 2$ | Yes | 22 k | 2.00 |
| Repeat-RNN, $\rho = 3$ | Yes | 49 k | 3.00 |
| Repeat-RNN, $\rho = 5$ | Yes | 27 k | 5.00 |
| Repeat-RNN, $\rho = 8$ | Yes | 26 k | 8.00 |

Table 2: Experimental results for the addition task.

| Model | Task solved | Training steps | Average repetitions |
|---|---|---|---|
| LSTM | No | - | 1.00 |
| ACT-LSTM, $\tau = 10^{-1}$ | No | - | 1.01 |
| ACT-LSTM, $\tau = 10^{-2}$ | Yes | 899 k | 5.08 |
| ACT-LSTM, $\tau = 5 \cdot 10^{-3}$ | Yes | 988 k | 6.74 |
| ACT-LSTM, $\tau = 10^{-3}$ | No | - | 11.91 |
| Repeat-LSTM, $\rho = 2$ | No | - | 2.00 |
| Repeat-LSTM, $\rho = 3$ | Yes | 997 k | 3.00 |
| Repeat-LSTM, $\rho = 5$ | Yes | 514 k | 5.00 |
| Repeat-LSTM, $\rho = 8$ | Yes | 576 k | 8.00 |

Figure 2 shows the impact of the amount of repetitions $\rho$ in Repeat-RNN for the addition task. We see that when increasing $\rho$ from 1 performance improves dramatically, but we also observe a limit: with too many repetitions the network shows instabilities with excessively long sequences. More figures with the evolution of the accuracy through training for both tasks and both models are included in Appendix B.

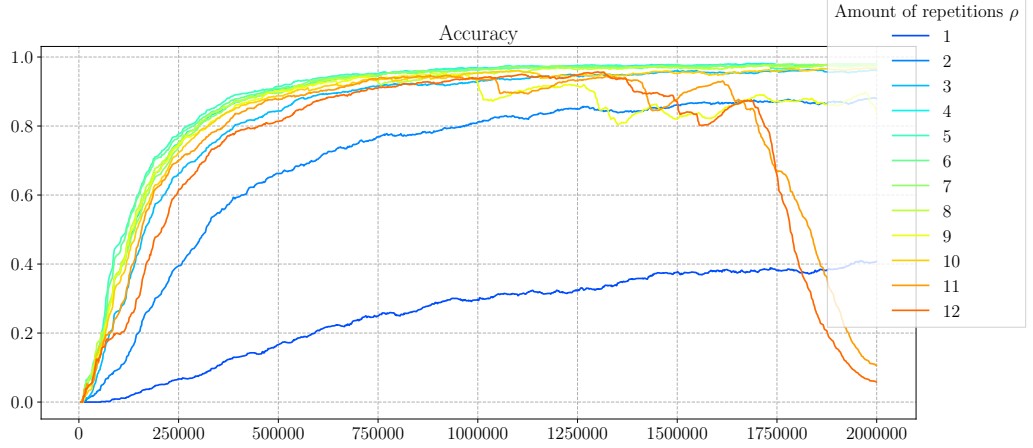

Figure 2: Accuracy for the addition task with Repeat-RNN.

## 4 DISCUSSION

We have introduced Repeat-RNN, a novel approach for processing each input sample for more than one recurrent step. This simple change is enough for a basic RNN or LSTM to be able to solve tasks such as parity or addition, respectively. We compared this new architecture to Adaptive Computation Time (Graves, 2016), which is also based on repeating inputs from a sequence, but does so dynamically. Surprisingly, Repeat-RNN performed as good or better than ACT in the considered tasks.

Repeat-RNN requires fixing a new hyperparameter $\rho$, the number of repetitions, which is task dependent. This tuning can be compared to the hyperparameter that must be fixed in ACT (the time penalty $\tau$), which is also task dependent and arguably less intuitive.

The reason why repeatedly updating the state for each input token before moving on to the next one improves the learning capability of the network remains an open question. The link between LSTM (Hochreiter & Schmidhuber, 1997) and Residual Networks (He et al., 2016), together with the recent finding that the latter perform iterative estimation (Greff et al., 2017; Jastrzebski et al., 2018), suggest that repeating inputs in RNNs may encourage iterative feature estimation and produce better performing models. Another possible reason is that the increased number of applied non-linearities allow the network to model more complex functions with the same number of parameters. Achieving a better understanding of the implications of repeating inputs remains as future work, and it may help in the design of better adaptive algorithms.

### ACKNOWLEDGEMENTS

This work was partially supported by the Spanish Ministry of Economy and Competitivity and the European Regional Development Fund (ERDF) under contracts TEC2016-75976-R and TIN2015-65316-P, by the BSC-CNS Severo Ochoa program SEV-2015-0493, and grant 2014-SGR-1051 by the Catalan Government. Víctor Campos was supported by Obra Social "la Caixa" through La Caixa-Severo Ochoa International Doctoral Fellowship program. We acknowledge the support of NVIDIA Corporation for the donation of GPUs.

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

# A    DESCRIPTION OF THE TASKS

## A.1    PARITY

The parity is not actually a sequence modeling task. A single input token is given to the network, which has to find the parity (or XOR) of the vector, in a single timestep. The input vectors has 64 elements, of which a random number from 1 to 64 is set to 1 or -1 and the rest are set to 0. The corresponding target is 1 if there is an odd number of ones, and 0 if there is an even number of ones. Each training sequence consists of a single input vector and a single target, which is a 1 or a 0.

The implemented network is single-layer RNN with 128 $\tanh$ units, and a single sigmoidal output unit. The loss function is binary cross-entropy and the batch size is 128. An example input and target are shown in figure 3.

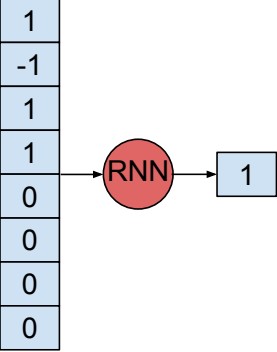

Figure 3: Input and target example of the parity task.

## A.2    ADDITION

The addition task aims at adding cumulatively the values of a sequence of five numbers, coded by one-hot encodings of their digits. Each number represented by $D$ digits, where $D$ is drawn randomly from 1 to 5. Each number is coded by a concatenation of $D$ one-hot encodings of its composing digits, each digit being value randomly chosen between 0 and 9. In case that $D$ is smaller than five, the representation of the number is completed with zero vectors, so that the total length of the representation per number is 50. The required output is the cumulative sum of all inputs up to the current one, represented as a set of six simultaneous classifications (for the 6 possible digits in the result of the sum). There is no target for the first vector in the sequence. Because the previous sum must be carried over by the network, this task requires the hidden state of the network to remain coherent.

Each classification is modeled by a size 11 softmax, where the first 10 classes are the possible digits and the 11[th] is a special marker used to indicate that the number is complete. An example input and target are shown in figure 4.

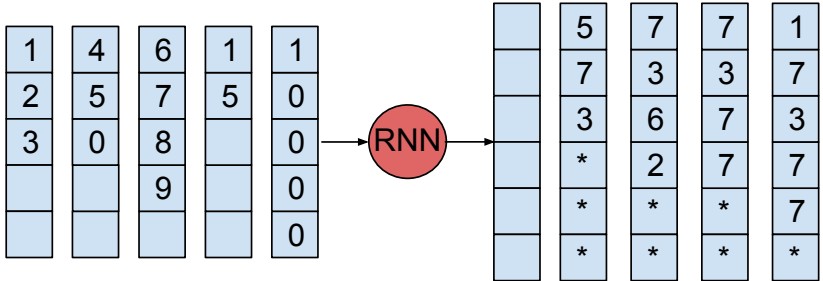

Figure 4: Input and target example of the addition task.

# B   FIGURES

For the parity task, figure 5 shows that a network with ACT is able to dramatically outperform a simple RNN and solves the task. On the other hand, figure 6 shows that Repeat-RNN is also able to solve the task when increasing the value of $\rho$ (the amount of repetitions). We can see that the performance of the network improves drastically with respect to $\rho = 1$, which corresponds to a simple RNN. We also observe that when doing to many repetitions, the network becomes unstable: even though it starts learning, the accuracy decreases in the late stages. We believe that this is might be caused by exploding gradients, because the sequence gets too long. For this task, it should be noted that an accuracy of 0.5 comes just from random guessing, since we are trying to predict 0's and 1's.

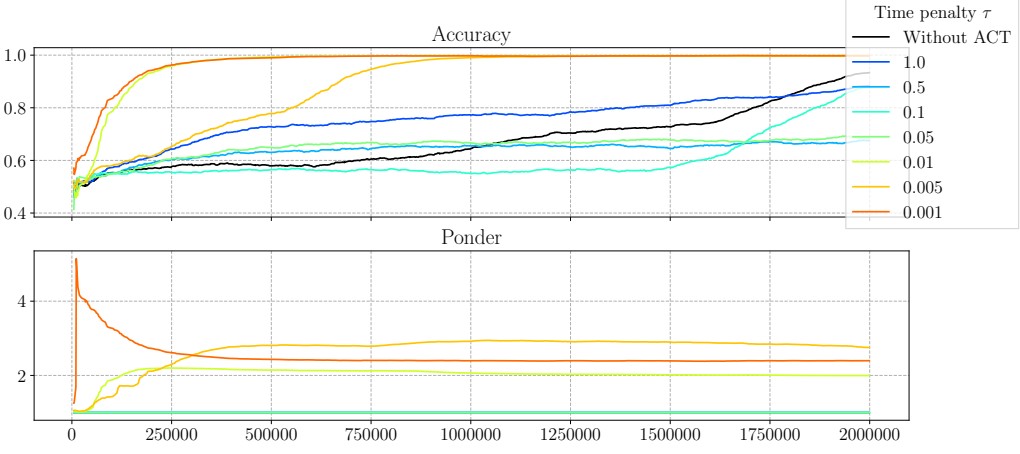

Figure 5: Accuracy and ponder cost for the parity task with ACT.

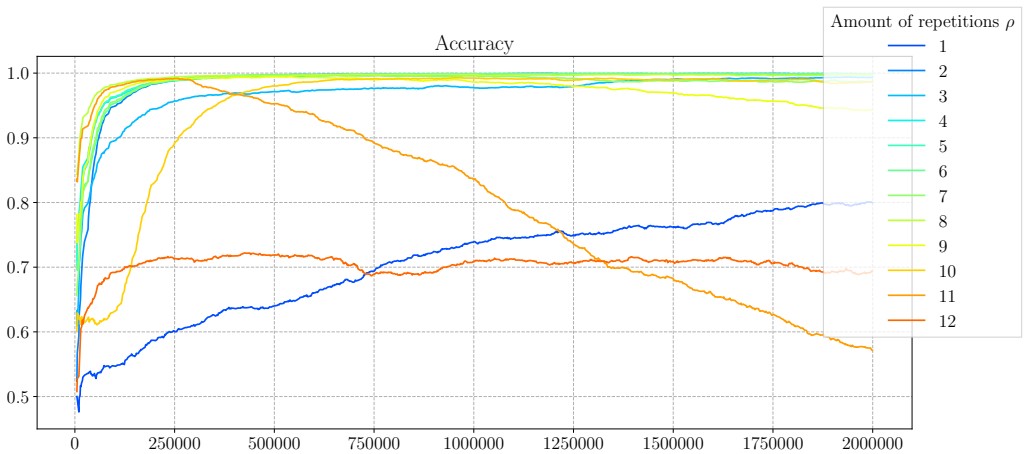

Figure 6: Accuracy for the parity task with Repeat-RNN.

For the addition task, in figure 7 we can see that ACT also increases performance over a simple RNN. Figure 8 shows the same task also being solved with Repeat-RNN. We see again that when increasing $\rho$ (amount of repetitions) we can also improve performance as much as ACT. We again see a limit: with too many repetitions we see the same problem as before, the networks shows instabilities with excessively long sequences.

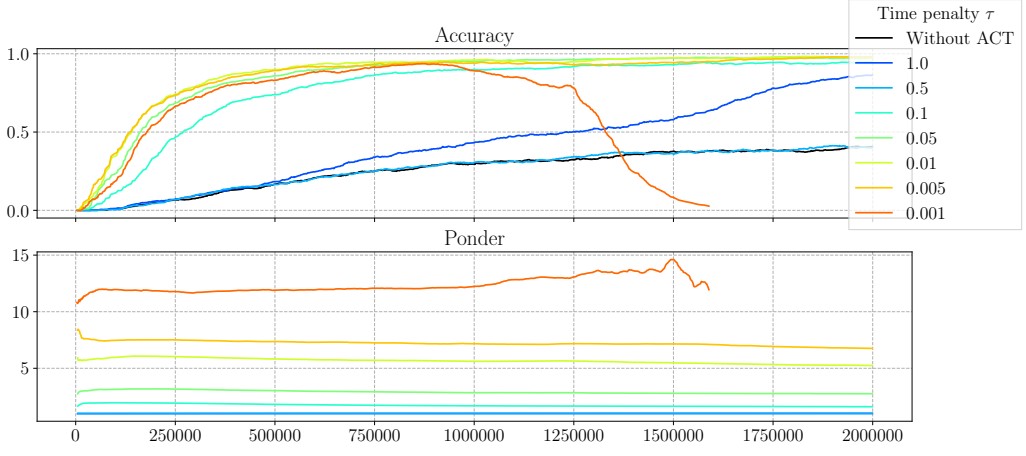

Figure 7: Accuracy and ponder cost for the addition task with ACT.

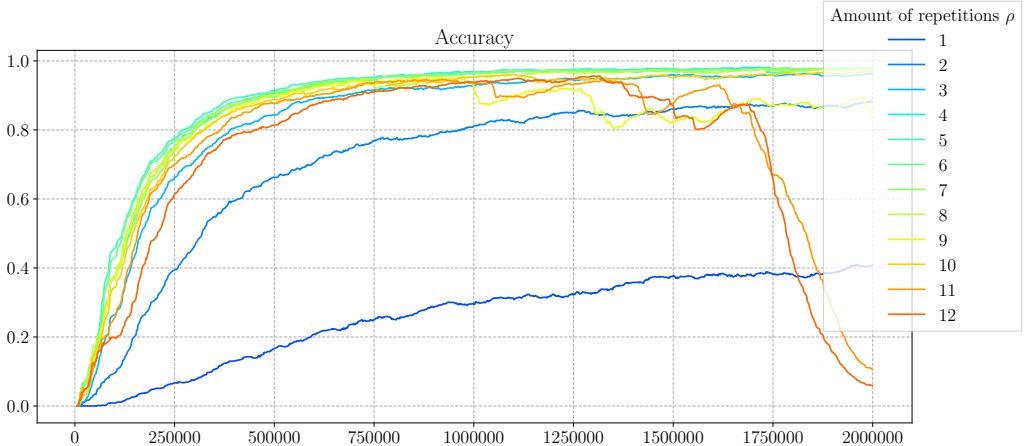

Figure 8: Accuracy for the addition task with Repeat-RNN.

