# OpenReview forum: "Comparing Fixed and Adaptive Computation Time for Recurrent Neural Networks"
_ICLR.cc/2018/Workshop — Accept_

### Official Review · AnonReviewer1 · 2018-03-09
**nice ablation test, more tasks would be great**

**Rating:** 7
**Confidence:** 4

**Review:**

The paper performs an ablation test for the Adaptive Computation Time model (ACT). Instead of learning to predict how many steps of computation are necessary for each input token, the proposed model (Repeat-RNN) performs a fixed number of steps. This number of steps is treated as a hyperparameter. Repeat-RNN is evaluated on two tasks from the ACT paper (parity and addition) and it is shown that it is better or not worse than in ACT in terms of both training and inference time.

The paper is clearly written, and I think that it makes a valuable contribution by  showing how a simple method can be just as good as a fancy one. I think we need such papers at the ICLR workshop.

A few suggestions for the authors:
- There was 5 tasks in the original ACT paper, and so far you only evaluated on 2. Would be great if you tried others as well.
- Consider citing https://arxiv.org/abs/1312.6026
- There is a bit of confusion in the text between notions of "input sample" and "input token", see e.g. item (1) in the intro
- item (1) in the intro has broken grammar

---

### Official Review · AnonReviewer2 · 2018-03-10
**Small demonstration that the important part of ACT is simply more steps.**

**Rating:** 5
**Confidence:** 5

**Review:**

The paper presents an ablation on the ACT (Graves 2016) architecture in which a constant number of ponder steps is performed between RNN steps. As in the original ACT, the ponder steps share weights.

Evaluation is done on two tasks from the original ACT paper - parity and addition, both tasks are not solved using a regular LSTM and require intermediate computation between RNN steps.

The idea of simply making the transition of an LSTM network more powerful is not new, as both adding more LSTM layers and more transitions (possibly with weight tying) was reported e.g. in http://proceedings.mlr.press/v70/zilly17a/zilly17a.pdf.

Pros:
- Very simple architecture

Cons:
- Limited novelty of the approach
- Lack of comparison with other approaches that add more compute steps, such as adding depth or the recurrent highway RNNs

---

### Official Review · AnonReviewer3 · 2018-03-12
**Ok, but more depth would be great**

**Rating:** 5
**Confidence:** 4

**Review:**

Your paper attempts to compare the speed and accuracy of an RNN when performing multiple recurrent steps for each input vs an adaptive number of steps for each input (ACT). This is generally OK but really not incredibly novel or difficult although some will find these limited results somewhat interesting -- however, many more experimental and theoretical directions could have been explored for a workshop paper.

Comments:
- Your title claims to compare fixed and adaptive computation time, however I don't see a single graph or table comparing computation times in sec, min etc.? Why not?
- The fact that running a fixed number of steps instead of an adaptive number of steps (which I think summarizes your contribution) and getting the same or better accuracy is to me (and I believe to many others) not surprising since you have more freedom when running more recurrent steps on average.
- I am surprised to read that updating the state repeatedly gives better accuracy remains an open question -- you are technically running through more layers which improves the modeling capacity, this is not really an open question.
- The E-Mail addresses and some of the graphs seem to go over the boundary of where text is allowed -- that probably has to be fixed.

---

### Public Comment · ~Oriol_Vinyals1 · 2018-02-17
**Formatting Violation**

Your paper violates the formatting posted here https://iclr.cc/Conferences/2018/CallForWorkshops. Please send us a fixed version of your PDF at iclr2018.programchairs@gmail.com by the end of Monday, February 19th, or else we will reject your paper.

Thanks,
ICLR2018 Program Chairs

---

> ### Public Comment · ~Daniel_Fojo1 · 2018-02-17
> **Re: Formatting Violation**
>
> We have sent the fixed version to the email, thank you.

---

### Decision · Program_Chairs · 2018-03-20
**ICLR 2018 Workshop Acceptance Decision**

**Decision:**

Accept

**Comment:**

Congratulations, your paper was accepted to the ICLR workshop.